# Single Cell ADNP Predictive of Human Muscle Disorders: Mouse Knockdown Results in Muscle Wasting

**DOI:** 10.3390/cells9102320

**Published:** 2020-10-19

**Authors:** Oxana Kapitansky, Gidon Karmon, Shlomo Sragovich, Adva Hadar, Meishar Shahoha, Iman Jaljuli, Lior Bikovski, Eliezer Giladi, Robert Palovics, Tal Iram, Illana Gozes

**Affiliations:** 1The Elton Laboratory for Molecular Neuroendocrinology, Department of Human Molecular Genetics and Biochemistry, Sackler Faculty of Medicine, Sagol School of Neuroscience and Adams Super Center for Brain Studies, Tel Aviv University, Tel Aviv 6997801, Israel; oxana188@hotmail.com (O.K.); gidikarmon@gmail.com (G.K.); srshlomo@gmail.com (S.S.); advaad@gmail.com (A.H.); elieze@tauex.tau.ac.il (E.G.); 2Department of Molecular Genetics, Weizmann Institute of Science, Rehovot 7610001, Israel; 3Intradepartmental Viral Infection Unit, Sagol School of Neuroscience, Tel Aviv University, Tel Aviv 6997801, Israel; meishars@mail.tau.ac.il; 4Department of Statistics and Operations Research, School of Mathematical Sciences, Raymond and Beverly Sackler Faculty of Exact Sciences, Tel Aviv University, Tel Aviv 6997801, Israel; jaljuli.iman@gmail.com; 5The Myers Neuro-Behavioral Core Facility, Sackler School of Medicine, Tel Aviv University, Tel Aviv 6997801, Israel; liorbiko@gmail.com; 6Department of Neurology and Neurological Sciences, Stanford University School of Medicine, Stanford, CA 95343, USA; palovics@stanford.edu (R.P.); tal.iram8@gmail.com (T.I.); 7Wu Tsai Neurosciences Institute, Stanford University School of Medicine, Stanford, CA 95343, USA

**Keywords:** muscular dystrophy, neuromuscular diseases, ADNP, NAP, CRISPR/Cas9

## Abstract

Activity-dependent neuroprotective protein (ADNP) mutations are linked with cognitive dysfunctions characterizing the autistic-like ADNP syndrome patients, who also suffer from delayed motor maturation. We thus hypothesized that ADNP is deregulated in versatile myopathies and that local ADNP muscle deficiency results in myopathy, treatable by the ADNP fragment NAP. Here, single-cell transcriptomics identified *ADNP* as a major constituent of the developing human muscle. *ADNP* transcript concentrations further predicted multiple human muscle diseases, with concentrations negatively correlated with the ADNP target interacting protein, microtubule end protein 1 (EB1). Reverting back to modeling at the single-cell level of the male mouse transcriptome, *Adnp* mRNA concentrations age-dependently correlated with motor disease as well as with sexual maturation gene transcripts, while *Adnp* expressing limb muscle cells significantly decreased with aging. Mouse *Adnp* heterozygous deficiency exhibited muscle microtubule reduction and myosin light chain (*Myl2*) deregulation coupled with motor dysfunction. CRISPR knockdown of adult gastrocnemius muscle Adnp in a Cas9 mouse resulted in treadmill (male) and gait (female) dysfunctions that were specifically ameliorated by treatment with the ADNP snippet, microtubule interacting, *Myl2*—regulating, NAP (CP201). Taken together, our studies provide new hope for personalized diagnosis/therapeutics in versatile myopathies.

## 1. Introduction

Essential for brain formation, activity-dependent neuroprotective protein (ADNP) [1,2,3] is a major brain regulatory gene [4]. *De novo* mutations in *ADNP* result in the *ADNP* syndrome, with children suffering from severe global developmental delays that affect the central and peripheral nervous systems [5]. Several structural abnormalities in the *ADNP* syndrome brain were observed including cerebral atrophy, delayed myelination, white matter lesions, and wide ventricles [5,6]. Other pathological outcomes include impairments in motor function, delayed speech acquisition, intestinal and urinary problems [5,7] as well as early tooth eruption [8]. Thus, *ADNP* mutations affect the three germ layers during development [9].

Using quantitative RT-PCR we have recently shown that *Adnp^+/−^* heterozygous deficiency in mice resulted in aberrant gastrocnemius muscle, tongue, and bladder gene expression, which was corrected by the ADNP fragment, drug candidate, NAP (CP201) [10]. A significant sexual dichotomy was revealed, coupled to muscle and age-specific gene regulation. Thus, Adnp regulated myosin light chain (*Myl*) transcript in the gastrocnemius muscle, the language acquisition gene forkhead box protein P2 (*Foxp2*) transcript in the tongue muscle, and the pituitary adenylate cyclase-activating polypeptide (PACAP) receptor PAC1 mRNA (Adcyap1r1) in the bladder, with PACAP, linked to bladder function [11] and partly controlling *Adnp* expression [12]. A sex-dependent *Adnp*-regulated gene transcript correlation to gait patterns (CatWalk) was discovered, placing ADNP as a muscle-regulating gene/protein [10].

We have also previously shown that the expression levels of *ADNP* and its paralogue protein *ADNP2* in the vastus laterlis and bicep brachii muscles are sex-dependently upregulated in elderlies compared to young subjects [13]. The *ADNP* transcript displayed age-dependent correlations with 49 muscle gene transcripts. Among these 49 transcripts, *ADNP* expression was highly correlated with 24 transcripts, singling nicotinamide nucleotide adenylyl (NAD) transferase 1 (*NMNAT1*) as the leading gene/protein [13]. Importantly, the NMNAT1-associated regulation of NAD^+^ salvage capacity in human skeletal muscle is declining with aging, while aerobic and resistance training attenuate this decline [14]. Thus, ADNP is implicated in human muscle aging.

A similar association of ADNP with aging has been suggested in the Alzheimer’s brain, with the observed correlation between somatic ADNP mutations and increased Alzheimer’s Tau pathology [13], directly linked to ADNP’s function as enhancing Tau-microtubule interaction [15].

Mechanistically, ADNP, a double-edged sword, is found in the cell nucleus and the cytoplasm [16]. In the nucleus, ADNP binds DNA, acting as a transcription factor [2,17], taking part in chromatin remodeling/epigenetic modifications [17,18,19] as well as RNA splicing interactions [20]. In the cytoplasm, ADNP binds the autism-linked eukaryotic initiation factor 4E (eIF4E) [21,22] and regulates microtubule dynamics by binding to the microtubule end binding proteins EB1 and EB3 [23], further enhancing Tau-microtubule interaction [15,24]. Microtubule function is sexually-dependent and *Adnp* deficiency results in reduced axonal transport [25]. The major cellular autophagy pathway is dependent on microtubule integrity [26], and ADNP also binds the microtubule-associated protein 1 light chain 3B (MAP1LC3B). Importantly, LC3 is a major protein forming the autophagosome [27]. ADNP binding to LC3 and EB1/EB3 is enhanced in the presence of NAP (NAPVSIPQ, drug candidate CP201) containing an EB1/EB3 and self-interacting SxIP motif [15,24,25,26,27].

Given the significant effect of ADNP mutations and ADNP deficiency on motor functions [28], we hypothesized that it is involved with versatile muscle diseases. Appendix A summarizes muscle diseases that may entwine with ADNP function including association with the cytoskeleton/microtubules, autophagy, and aberrant gene regulation. In short, these diseases may include myotonic dystrophy (e.g., DM2) [29], Duchenne muscular dystrophy (DMD) [30,31], Becker muscular dystrophy (BMD) [32,33], Pompe disease [34], tibial muscular dystrophy (TMD) [35], dysferlinopathy [36], secondary dystroglycanopathies [37] and amyotrophic lateral sclerosis (ALS) [38].

Taking advantage of single-cell transcriptomic data in humans and mice, we were able to show the developmental expression of ADNP in human muscle progenitor cells and correlate muscle disease genes with *Adnp* in mice. Modeling *Adnp* deficiency in mice, we further asked whether such deficiency was linked to muscle regulation. Lastly, we investigated if Cas9-mediated muscle-specific adult mouse Adnp knockdown resulted in motor deficiencies, which could be significantly reversed by NAP treatment.

## 2. Materials and Methods

### 2.1. Human Muscle Single Cell Data Mining

The NCBI GEO website was screened for datasets derived from human single-cell transcriptomes of skeletal muscles. In the dataset GSE147457 [39] human tissues including hind limbs of developmental weeks 5–18 human embryos and fetuses, and gastrocnemius or quadriceps muscles from juvenile and adult human were processed into single cells. The UCSC Cell Browser [40] was used to analyze the data. 2D plots of single cells were visualized by the t-SNE algorithm, and ADNP expressing cells were marked in black. The expression levels of ADNP in developmental weeks 5–18, years 7, 11, 34, and 42 were downloaded from UCSC single-cell browser. Average expression levels for each cell type and time point were calculated, and the dot plot figure was generated with Tableau 2019.2 software. The Violin plots were generated for cells expressing ADNP, DMD, GAA, and DYSF with ggplot, RStudio Version 1.2.5033 (RStudio, PBC, Boston, MA, USA). 

### 2.2. Online Gene Expression Omnibus (GEO) Public Functional Genomics Data Repository, Human Muscle Diseases

A summary of the different patients and age-matched controls is presented in Appendix A, Specific datasets included GAA, GSE38680 [41], GSE1007 [42], GSE45331 [43], GSE42806 [43] and GSE3307 [44,45].

### 2.3. Secondary Analysis of Tabula Muris Senis Single-Cell Data Set

Raw cell-to-gene count matrix along with sample metadata was acquired from AWS using project ID arn:aws:s3:::czb-tabula-7 muris-senis. Here, the FACS Smart-seq2 data uploaded on 12 April 2019 was analyzed. To minimize possible sources of bias, female cells, which had lower coverage across cell types and time points, were excluded from subsequent analysis. Cells with either less than 500 genes detected or 50,000 reads in total were removed from the expression matrix. Counts were then normalized as log(CPM + 1). To visualize the cell-type clusters we computed t-SNE embeddings over the 16 principal components of the normalized expression matrix. The similarity between the expression of all detected genes and Adnp was calculated based on the cosine similarity. All computational steps were done by using Python with the packages Scanpy v1.4.4, Pandas v1.0.1, Numpy v1.18.1, and Scikit-learn v0.22.1 (DataCamp, New York, NY, USA).

### 2.4. Animals

All procedures were approved by the IACUC of Tel Aviv University and the Israeli Ministry of Health (01-18-020; 01-18-018).

Two mouse models were used:

(1) The *Adnp^+/−^* mice, on a mixed C57BL and 129/Sv background [4,21,28,46,47], outbred with an ICR mouse line for continuous breeding [21,25].

(2) Cas9 expressing Gt(ROSA)26Sor^tm1.1(CAG-cas9*, -EGFP) Fezh^|J (Jackson Laboratory, stock: 024858) males crossbred with female ICR mice and progeny subjected to muscle *Adnp* knockdown (Cas9 mice, below).

#### 2.4.1. Neuromuscular Junction (NMJ) Staining

Medial gastrocnemius muscles were excised from 7-month-old and 14-month-old *Adnp^+/−^* male mice and stained as before [48,49]. Digital images were obtained using a Leica SP5 confocal laser scanning microscope [48].

#### 2.4.2. Peptide Synthesis and Formulations

NAP was custom synthesized and formulated as before [28,50,51,52]. Specifically, NAP was dissolved in a vehicle solution termed DD, in which each milliliter included 7.5 mg of NaCl, 1.7 mg of citric acid monohydrate, 3 mg of disodium phosphate dihydrate, and 0.2 mg of benzalkonium chloride solution (50%). A second vehicle solution termed CB included 0.25% chlorobutanol, 0.85% NaCl, pH = 3.5 to 4.0. On days of scheduled behavioral tests, NAP was applied 2 h before the behavioral tests.

#### 2.4.3. RNA Extraction and Quantitative Real-Time PCR

*Adnp^+/−^* mice were divided into groups and sacrificed for RNA extraction by the end of the treatment period. All behavioral assessments and RNA extractions (TRI Reagent, T9424, Sigma-Aldrich, Jerusalem, Israel) and gene expression analysis were carried out before as described before in a semi-blinded manner [28,51] and correlated with gene expression results. RNA expression levels were determined using specific mouse primers: *Adnp* sense 5′-ACGAAAAATCAGGACTATCGG-3′, anti-sense 5′-GGACATTCCGGAAATGACTTT-3′, *Myl2* sense 5′-GCCCTAGGACGAGTGAA-3′, anti-sense 5′-CCAAACATCGTGAGGAAC-3′. mRNA levels were normalized to *Hprt* sense 5′-GGATTTGAATCACGTTTGTGTC-3′, anti-sense 5′-AACTTGCGCTCATCTTAGGC-3′. Results are presented as 2^−ΔCT^ [53].

Muscle *Adnp* knockdown Cas9 mice were treated daily throughout the experiment (0.5 μg intranasal NAP/5 μL DD per mouse) and subjected to behavioral experiments (below).

#### 2.4.4. Virally-Delivered CRISPR-Mediated Adnp Knockdown in Muscle Tissue of Cas9 Mice: Cell Lines for Reagent Preparations

NIH 3T3 cells, mouse fibroblasts, and HEK-293T, human embryonic kidney cell lines (ATCC), were plated in DMEM supplemented with 10% fetal calf serum, 2mM glutamine, and 1% penicillin-streptomycin (Biological Industries, Beit HaEmek, Israel) [17]. The cells were incubated in 95% air/5% CO_2_ in a humidified incubator at 37 °C.

#### 2.4.5. Single Guide RNA (sgRNA) Preparation and Plasmid Construction

Three different sgRNA were designed using CRISPOR [54]. To determine the effectivity of the designed *Adnp* sgRNAs (targeting the second coding exon (exon No.4) of Adnp), NIH 3T3 cells (ATCC) were transfected in triplicates with PX459 plasmids (Addgene #62988, from Feng Zhang, Broad Institute, MIT and Harvard, Cambridge, MA, USA) [55], encoding three different *Adnp* sgRNA (termed 60,67,68) and an empty plasmid as a control using jetPEI (101-10N, Polyplus Transfection, Illkirch, France). Transfected cells were selected using 3 ug/mL Puromycin (P7255, Sigma-Aldrich) for two weeks. Extracted total cellular protein was subjected to Western blotting [24] and ImageJ software quantification (NIH, Bethesda, MD, USA) [15].

#### 2.4.6. Lentivirus Production and Injection

Lentiviral vectors were generated by insertion of the guide-RNA sequences, SgRNA 68 and a stuffer (poly T), into the third generation lenti backbone pAW13.lentiguide.mCherry (Addgene #104375, a gift from Richard Young, MIT, Cambridge, MA, USA) [56]. High-titer lentiviral stocks pseudo-typed with the vesicular stomatitis virus G protein (VSV-G) were produced in HEK293T cells [57,58]. 3-month-old Cas9-expressing male and female mice were anesthetized using isoflurane. Next, 100 μL of Neurobasal medium containing lentiviruses at a concentration of 2.96 × 10^7^ titer units, expressing either G68 (knocking down the *Adnp* gene) or Poly T (stuffer) were injected into gastrocnemius muscles, using a 1 mL syringe and a 25G needle. Each virus was injected into both hind limbs of the same animal. Animal group sizes were determined in a pilot study. Cas9 mice from five litters were included in the study and randomly divided into four different groups, per sex as follows: (1) Injection control group (NB) (male *n* = 4; female *n* = 4), (2) PolyT DD (male *n* = 12; female *n* = 8), (3) G68 DD (male *n* = 7; female *n* = 7), (4) G68 NAP (male *n* = 8; female *n* = 7).

#### 2.4.7. Behavioral Studies

A month following viral injections and NAP or DD daily treatments, male and female Cas9 mice were subjected to multiple behavioral tests.

#### 2.4.8. Gait Analysis

CatWalk XT (Noldus Information Technology, Wageningen, The Netherlands) was used [28,59,60].

#### 2.4.9. Treadmill

The treadmill apparatus (Panlab, Harvard Apparatus, Barcelona, Spain) was used [61].

#### 2.4.10. Hot plate

The hot plate (Ugo Basile, Gemonio VA, Italy) was used [62].

### 2.5. Statistics

Results were analyzed for statistical significance using (1) Sigma Plot for Windows software version 11 (Chicago, IL, USA) and (2) R Core Team (2019) for Mac, version 3.6.2 (2019-12-12) (R Foundation for Statistical Computing, Vienna, Austria). The analyses and graphs produced by R were performed using the libraries: lme4, lmerTest, and ggplot2. The reported estimators of effects and differences in pairwise comparisons were obtained by the Restricted Maximum Likelihood (ReML) method.

The models used for data analyses were one- or two-way ANOVA and logistic regression mixed-effects as detailed below. Statistical significance of the pairwise comparisons was evaluated ty the model-corresponding post-hoc t-test, with correction multiple comparisons. Sex differences were revealed between male and female mice using an unpaired Student’s t-test.

Inclusion/exclusion of values per each tested group were performed using Grubbs’ test. All determinations were considered to be statistically significant at the *p* < 0.05 level: * *p* < 0.05, ** *p* < 0.01, *** *p* < 0.001.

## 3. Results

### 3.1. Human Single Cell ADNP Is Increased in Single Muscle Progenitor Cells

To place ADNP transcript expression directly in human muscle cells, we resorted to whole-cell transcriptomics. We used an available dataset (GSE147457). As previously published, whole hind limbs of developmental week 5–9 human embryos and fetuses (feet excluded for week 7.75–9), total hind limb skeletal muscles of week 12–18 human fetuses, and gastrocnemius or quadriceps muscles from juvenile and adult human subjects were digested into single cells. Dissociated cells were either sorted (fetal week 12 and above) to exclude the hematopoietic and endothelial lineages or directly used for downstream processing [39]. Figure 1A shows single-cell *ADNP* transcriptomes of skeletal muscles, gastrocnemius, and quadriceps muscles from human specimen ranging from 5-week-gestation to 42 years of age. Results indicated increased expression in the young embryonic muscle. Figure 1B focuses on 9-day gestation hind limb cells indicating enrichment of *ADNP* in pre-chondrocytes and skeletal muscle cells, showing expression in dividing cells of the myogenic subset myogenic progenitor cells (inset). Appendix A details cellular expression at 12–14 and 17–18 weeks of gestation, respectably, with skeletal muscle at all stages, and with mesenchymal stromal cells increasing at 12–14-gestational-weeks and further increasing at 17–18-gestational-weeks.

### 3.2. ADNP Expression Plays a Role in Childhood and Adult-Onset of Neuromuscular Disorders

Figure 1C,D depict violin single-cell expression graphs at the 9-gestational week of ADNP and dystrophin (DMD), a major neuromuscular disease gene, as an example (Appendix A), indicating some similarities. Given the potential parallelism between ADNP and the expression of DMD as well as other gene transcripts representatives of muscle disease (Appendix A), it was of interest to address the question if *ADNP* expression is altered in neuromuscular disorders. Thus, the online Gene Expression Omnibus (GEO) public functional genomics data repository was searched for expression datasets of human subjects afflicted with various neuromuscular disorders, comparing transcriptional profiles of controls and affected individuals. A summary of the different patients and age-matched controls is presented in Appendix A, indicating specific muscle sampled and sex. Interestingly, microarray expression levels of *ADNP* in different muscle types showed significant downregulation in four out of the eight tested neuromuscular disorders including Pompe, caused by mutations in the acid alpha-glucosidase (GAA, GSE38680) [41] (sampled tissue, biceps), Duchenne muscular dystrophy caused by dystrophin (DMD) absence (GSE1007) [42], (quadriceps-vastus lateralis), myotonic dystrophy type 2 (DM2), caused by mutations in the cellular nucleic acid-binding protein gene, CNBP (GSE45331) [43] (vastus lateralis), and tibial muscular dystrophy (TMD) caused by mutations in titin (TTN, GSE42806) [43] (distal muscles), compared with healthy controls (Figure 2A–D). Further analysis of the GEO data set GSE3307 [44,45] showed significant increases in the expression levels of ADNP in the vastus lateralis of other muscle pathologies including Becker muscular dystrophy (BMD, DMD mutations), dysferlin mutation (DYSF) [63], fukutin-related protein (FKRP) mutation [64] and amyotrophic lateral sclerosis (ALS), compared with matched healthy controls (Figure 2E,F). We were also interested in whether each of these eight diseases can be anticipated by *ADNP* expression levels. In Pompe, DM2, TMD, and BMD patients, *ADNP* expression values essentially did not intersect with control values; therefore, the rule of differentiation is quite clear, not requiring statistical modeling. For the other tested diseases (DMD, FKRP, DYSF, and ALS) with extensive overlapping values, we performed logistic regressions estimating the probability of disease diagnoses via *ADNP* levels. Results showed significantly predictive values (Figure 2G). The tendency to develop each of the illnesses, for any given expression level was predicted by logistic regression. To draw a differentiating line between expression levels with high risk, we used the default probability cutoff, 0.5. Explicitly, expression levels that were accompanied by the probability of illness that was over 50% (dashed vertical lines) were classified as indicators of future diagnosis (Figure 2G).

Given the dichotomous downregulation versus the upregulation of *ADNP* in the different diseases, we examined also the ADNP cytoplasmic targets EB1 and EB3 also called microtubule-associated protein RP/EB family member 1 (MAPRE1) and MAPRE3. We thus discovered that *MAPRE1* and to a lesser extent *MAPRE3* are regulated in an opposite way compared to disease-downregulated *ADNP* (Figure 2A–C). Specifically, both *MAPRE1* and *MAPRE3* were upregulated in Pompe disease (Figure 2A), and *MAPRE1* was upregulated in DM2, TMD, and DMD (Figure 2B,C,H). In contrast, in ALS and BMD, *MAPRE3* expression did not differ between control and disease, and *MAPRE1* was increased (Figure 2H). In DYSF and FKRP, no significant differences were found for *MAPRE1* or *MAPRE3* between controls and disease afflicted subjects. Individual correlations showed a highly significant negative correlation between *MAPRE3* and *ADNP* expression in Pompe disease (Figure 2A, r = −0.781, *** *p* < 0.001, Spearman). A similar correlation was found in TMD for *MAPRE1* and *ADNP* (Figure 2C, r = −0.620, **p* = 0.0315, Pearson). Lastly, in ALS, a positive correlation was discovered between the expression of *MAPRE1* and *ADNP* (Figure 2F,H r = +0.601, *** *p* < 0.001, Spearman).

### 3.3. Mouse Single Cell Analysis Age-Dependently Correlates Adnp to Muscle Disease Genes

Although correlation is not causation, we were interested in correlating *Adnp* gene transcript concentrations to muscle disease genes at the single-cell level [13,65,66]. We resorted to data mining of two libraries, the young (3-month-old) and the aged (18 and 24 months) male mice, and focused on 8 genes as illustrated in Table 1. For each of the genes, we showed the cell type exhibiting the highest correlation to *Adnp* (see methods), in young mice, and further showed that these correlations were mostly lost in aged mice. Surprisingly, the tissue origin of the cells showing the highest correlative values were the brown adipose tissue, the pancreas, the large intestine, and the lung, possibly implicating *Adnp* and muscle disease genes in functions beyond the neuromuscular junction (NMJ). Only *Fkrp* (associated with adherence to the extracellular matrix, Appendix A) remained correlated with *Adnp* in the aging bronchial smooth muscle cell.

At the general limb muscle level, most of the disease genes in which *ADNP* reduction was disease predictive correlated with *Adnp* expression in the young specimens. *Cnbp* seemed relatively highly correlated with *Adnp*, associated with the most common adult myotonic dystrophy, DM2, (Appendix A), while *Ttn* showed almost no correlation (Figure 3). Of note, for ALS, we have used *C9orf72*, which most certainly does not represent all cases of sporadic ALS.

Furthermore, genes in which reduction in *ADNP* was disease predictive (Figure 2), showed apparent higher *Adnp* correlations compared to genes in which increases in *ADNP* correlated with the disease (Figure 3 vs. Appendix A, respectively). The correlations were reduced with aging (Figure 3) with disease genes associated with *ADNP* increases, completely losing correlation with aging (Appendix A).

In general, searching for genes with the highest correlative expression to *Adnp* at the cellular level, we discovered in the young male mouse (and not in the aged mouse), a 0.9 correlation with the gonadotropin-releasing hormone receptor (*Gnrhr*) transcript in diaphragm mesenchymal stem cells, tying *Adnp* to sexual maturation, with muscle diseases being highly sexually dichotomized (Appendix A).

Overall cell-specific age and sex-dependent expression (male) patterns of *Adnp* were observed (Figure 4A), with significant differences marked with asterisks. In most tissue-representing cells, *Adnp* expression, and/or the fraction of cells expressing *Adnp* significantly decreased with aging, except for liver, aorta, and mesenchymal fat, showing low expression levels (Figure 4A). To evaluate mouse *Adnp* age-dependent expression in limb muscle cell types (Figure 4B), we specifically looked at the single-cell level where *Adnp* is found in various limb muscle cell types (see blue dots in Figure 4C) with a significant decrease in *Adnp* positive cells remaining in aged muscle (red dots in Figure 4D).

### 3.4. Adnp^+/−^ Mice Display Neuromuscular Junction (NMJ) Disruption, Significantly Correlated with Behavioral Deficits

To further evaluate the mechanistic role of ADNP in muscle function, we utilized muscle histochemistry and gene expression measurements in *Adnp^+/−^* mice.

*Adnp^+/−^* mice treated with the intranasal formulation chlorobutanol (CB), serving as controls for future drug studies (employing the CB formulation), were previously used [51]. Here, representative whole-mount NMJ staining of the gastrocnemius muscle obtained from 7-month-old CB-treated *Adnp^+/+^* and *Adnp^+/−^* male mice are shown (Figure 5A). Sections were labeled with the post-synaptic marker α-bungarotoxin (BTX) and the pre-synaptic marker tubulin (TUB 2.1 monoclonal antibodies) [48,49,51]. Results indicated significantly decreased tubulin intensities in *Adnp^+/−^* vs. *Adnp^+/+^* muscles (Figure 5A, reduction of 21.5%). We have repeated and extended this result in muscles from 14-month-old mice showing a 32.3%-genotype related reduction in tubulin staining (Appendix A).

Quantitative real-time PCR (qRT-PCR) assays evaluating *Adnp* mRNA expression were further performed on RNA extracted from gastrocnemius muscles of CB-treated 7-month-old male and female mice (*Adnp^+/+^* and *Adnp^+/−^*). A significant genotype-related reduction was observed in males (Figure 5B), but not in females. Most importantly, significant positive correlations were discovered between behavioral test results (hanging wire, measuring the latency to fall off an inverted cage lid [51]) and gene expression levels (*Adnp*) in muscles of 7-month-old mice (Figure 5C). Further hanging wire behavioral correlations were discovered with myosin light gene 2 (*Myl2*), a major gene regulated by Adnp [9,25] in a sex-dependent manner (Figure 5D,E).

To further substantiate the mechanistic basis for the effects of ADNP in relation to Myl2 on muscle development/function, we examined the water-based DD-formulated/NAP-treated *Adnp^+/–^* and *Adnp^+/+^* mice, extensively studied for brain function and behavior before (methods) [28]. Thus, we assessed *Adnp* and *Myl2* gene expression patterns in the gastrocnemius muscle in correlation with behavior in 19–27-days-old mice.

*Adnp* reduction as a consequence of *Adnp* gene copy deficiency (*Adnp^+/−^*) was significant in both males and females, coupled with higher expression of *Adnp^+/+^* males compared with females. *Myl2* showed the opposite expression pattern, being higher in *Adnp^+/+^* females, significantly reducing with *Adnp* deficiency in females only, and normalized by the ADNP snippet NAP treatment (Figure 6A).

We also correlated *Adnp* and *Myl2* expression levels with previously published behavioral outcomes [28] and significant results are depicted in the Table insert in Figure 6A. In short, both *Adnp* and *Myl2* highly correlated with the first day of acquisition of the negative geotaxis response in females. *Adnp* expression was correlated in a sex-dependent manner with CatWalk gait measurements [10] (front paws, males only and hind left paw, males and females). As expected from the pattern of gene expression, *Myl2* correlated with CatWalk gait measurements in females only in both hind paws.

### 3.5. ADNP is Functionally Associated with Multiple Muscle Disease Proteins

Given the tight association of *ADNP* expression levels with key motor dysfunctions, we asked if the mutated proteins causing these diseases (Appendix A) are linked with ADNP at the protein level. STRING analysis for functional interactions included the mutated muscle disease-causing proteins (Figure 6B) alpha-actin (ACTA1), a major cytoskeletal gene, MAP1LC3B, a protein regulating autophagy, directly binding ADNP [13,27] and for ALS, C9orf72, a gene also linked to autophagy [68] and interacting with CNBP (causing DM2). In this respect, CNBP interacts with the α subunit of the dystroglycan complex, a core component of the multimeric dystrophin-glycoprotein complex, which regulates membrane stability [69]. As predicted, our results identified a tight network of proteins associated with cytoskeleton and muscle function (Figure 6B). Importantly, muscle *Myl2*, discovered here as not only regulated by ADNP in a sex-dependent manner but also corrected by NAP treatment, was seen as an integral part of the protein network regulating muscle function. Indeed, MYL2 has an obvious association with myosin’s essential role in muscle contraction.

### 3.6. Choosing the Most Efficient Guide RNA for Muscle Adnp Knockdown

To provide direct evidence for the involvement of Adnp in adult muscle function, we virally-delivered the CRISPR-mediated knockdown of Adnp to the gastrocnemius muscle of Cas9 adult mice. Specifically, we designed three sgRNAs (G60, G67, and G68) to target the second coding exon (exon No.4) of the mouse *Adnp* gene (Figure 7A). To quantitatively determine the relative efficiency of the designed sgRNAs, we transfected these sgRNAs and the Cas9 protein into NIH 3T3 cells (of murine origin), and after two weeks evaluated the Adnp protein levels (Figure 7B and Appendix A). All three sgRNAs showed a significant reduction in Adnp protein levels, with G68 showing the most profound knockdown (98.65% reduction) compared with the appropriate control (Figure 7B). The off-target effect of G68 (listed in Appendix A) has been proven to be most likely negligible. Thus, lentiviruses with sgRNA G68 and a stuffer (a stretch of poly T causing premature termination of RNA pol III as a control [70]) were prepared as described before [58] and injected into gastrocnemius muscles of 3-month-old Cas9-expressing mice. One-month post-injection, during which animals were treated with intranasal NAP or vehicle (DD, Methods), we performed motor behavioral experiments including the treadmill test and the CatWalk gait analyses.

### 3.7. Adult Adnp Knockdown Male Mice Exhibit Aberrant Motor Performance in The Treadmill Test

In the treadmill test (Figure 7C), we assessed: total distance run, time until exhaustion, and maximum speed. The maximum running speed was significantly lower in G68 DD male mice, compared with Poly T DD, and G68 NAP treatment ameliorated this deficit (Figure 7C). Furthermore, sex differences were found in the G68 NAP-treated group (Figure 7C and Appendix A), with the female group showing reduced performance in all tested parameters, compared with male mice. Additionally, G68 DD male mice remained on the treadmill for significantly shorter periods of time compared with the Poly T DD group (Appendix A), conclusively running shorter distances (Appendix A). NAP-treated G68 male mice exhibited a trend of improvement (*p* = 0.053) in time until exhaustion, as well as in running performance (*p* = 0.06) (Appendix A, respectively). Importantly, we also performed a hot plate test to exclude the possibility that the phenotype observed in the G68 mice was a consequence of sensory deficits caused by the lentiviral injection procedure. We did not observe any significant changes between the tested groups (Figure 7D).

### 3.8. Impaired Gait Parameters in Cas9 Female Mice Adnp Knockdown Are Ameliorated by NAP Treatment

Using the Catwalk apparatus, we estimated a substantial number of gait parameters divided into several categories: run characterization, interlimb coordination (swing speed, body speed, step cycle, and base of support (BOS)), as well as temporal and spatial parameters. (Our previous paper [10] details the precise measurement characteristics).

Figure 8 depicts an extensive analysis of gait parameters that were affected by *Adnp* knockdown and corrected by NAP treatment. Interestingly, those changes were discovered in females, whereas males were mostly unaffected.

Specifically, results showed that swing speed was significantly decreased in G68 DD female mice, as compared with the Poly T DD group and this decrease was ameliorated by NAP treatment (Figure 8A). Correspondingly, body speed displayed a similar pattern (Figure 8B). Our analysis also revealed a significant increase in the duration of the step cycle in G68 DD female mice, as compared with Poly T DD and NAP-treated G68 groups (Figure 8C).

Another coordination-related parameter analyzed was the BOS. Here, we observed a significant decrease in BOS (hind paws) in the G68 DD female group in relation to the Poly T DD and the NAP-treated G68 mice (Figure 8D).

A significant difference between Poly T DD and G68 DD mice was also found in temporal parameters. Single stance, was significantly higher in the RH G68 DD females, compared with Poly T DD and NAP-treated (ameliorated) G68 DD mice. This was the only parameter ameliorated by NAP treatment, which showed an increase because of *Adnp* knockdown and an effect on the right hind paw (Figure 8E).

The spatial parameters were also altered in G68 DD female mice (Figure 8F). G68 DD female mice showed a significantly reduced LF paw contact pressure with the walkway, compared with the Poly T DD group, measured as the mean intensity of the 15 most intense pixels and corrected by NAP treatment (Figure 8F). Sex differences were also observed in the G68 DD group (***p* < 0.01). An additional summary of significant changes of the affected parameters by either the *Adnp* genotype, or the NAP treatment or by sex in CatWalk gait results is presented in Appendix A, showing multiple small, but significant differences.

To exclude the possibility that the observed phenotype in G68 mice was attributed to the injection procedure, we used an additional mouse group injected only with the carrier Neurobasal (NB) medium. Appendix A shows no significant differences between the Poly T DD and NB groups, strengthening the above results.

## 4. Discussion

Our paper investigated the involvement of *Adnp* expression and NAP ameliorative effects in developing/adult muscle function at four levels: (A) the single cell, (B) human muscle diseases, (C) inborn *Adnp* deficiency (*Adnp^+/−^* mice), and (D) adult muscle genome editing, Adnp knockdown. Together, *ADNP/Adnp* was identified in the single developing/aging muscle cell. *ADNP* expression presented predictive values to human muscle diseases. The mechanistic mouse results implicated *Adnp* regulation of microtubules and myosin (*Myl2*) as key players regulating behavioral outcomes in a sex/age-dependent manner. Importantly, direct Adnp local knockdown exerted sex-specific muscle defects, which were partially ameliorated by intranasal NAP treatment.

Given our results correlating between *ADNP* expression and human muscle aging and our current transcriptomic analyses at the single human muscle cell level indicating a potentially crucial role for ADNP in muscle development, as exemplified in the ADNP syndrome [5,7,8,71], we investigated the involvement of ADNP in several NMJ pathologies (some highly prevalent in males) and discovered significant dysregulation. A Comparison of *ADNP* levels in Pompe [41], DMD [42], DM2 [43], and TMD [43] muscles with healthy controls revealed highly significant (essentially nonintersecting) downregulation compared with control values. DMD presented an exception with some overlap between disease and control individuals, regardless, also in the DMD case, ADNP levels were predictive of the disease. In this respect, Pompe and TMD are linked with autophagy (Appendix A), which is in turn linked with ADNP [26,27]. DM2 is associated with aberrant RNA splicing, also linked with ADNP function [20,72]. DMD (dystrophin absence) is linked to the cytoskeleton, with dystrophin binding to actin [73] and actin filaments interacting with microtubules [74,75].

In BMD (dystrophin mutation), dysferlin mutation (DYSF, associated with mitochondrial function) [63], fukutin-related protein mutation (FKRP, associated with extracellular matrix) [64] and ALS (Appendix A), *ADNP* transcript levels, although predictive of the pathology, not showing complete separation, increased compared with matched healthy controls. These increases may represent an age/sex/muscle-dependent compensatory effect, also resulting from ADNP auto-regulation of its own gene expression [9,76]. Regardless, in a zebrafish model of *FKRP* mutation, NAD^+^ supplementation prior to muscle development improved muscle structure, myotendinous junction structure, and muscle function [64], with *ADNP* highly correlated with *NMNAT1* (e.g., the aging human muscle [13]). Furthermore, in the SOD1-G93A mouse model of ALS, we have shown an involvement of microtubule-tau pathology and protection by NAP [77] (Appendix A). These results also connect with the extensive interactions of cellular cytoskeletal elements affecting microtubule dynamics [78], with ADNP/NAP playing an essential role in maintaining microtubule dynamicity and regulating multiple gene expression patterns, in a sex-dependent manner [25]. Importantly, our functional analysis of muscle disease protein interactions (Figure 6) tightly linked ADNP with human muscle contraction and implicated direct ADNP interactions with muscle motor disease inflicting proteins. These interactions are intimately interwoven with our findings of decreased microtubules in the limb muscle of the *Adnp^+/−^* mice, which was exacerbated by aging. We suggest that deficits in the ADNP/microtubule cytoskeletal network may increase the susceptibility of the NMJ to physical disruption. Such alterations in the microtubule network density would have a profound impact on the contractility of the *Adnp^+/−^* muscle. Thus, our results imply an important role for ADNP in maintaining proper NMJ structure and function and as emphasized above, with STRING pathway analysis strongly linking ADNP with key muscle disease proteins, at the level of cytoskeletal protein organization.

To further understand the differential involvement of ADNP in the various muscle disorders, we have also assessed the levels of the ADNP binding proteins EB1 (MAPRE1) and EB3 (MAPRE3). Our results indicated that in all the diseases showing *ADNP* decreases, *MAPRE1* increased, and Pompe disease even showed an additional increase in *MAPRE3*, potentially compensating for the apparent ADNP deficiency. In contrast, the increase of *ADNP* in BMD and ALS was coupled to an increase in *MAPRE1*, while DYSF- and FKRP-related diseases did not show a significant change in *MAPRE1*. These results suggested differential regulation of *ADNP/MAPRE1* in the different muscle diseases, with *MAPRE1* consistently increasing (or showing no significant change) and with *ADNP* auto-regulating its own transcript levels [9,76] in a feedback mechanism. Regardless, as indicated above, in an ALS mouse model, NAP provided protection [77]. Furthermore, attesting to the importance of the *MAPRE1,3* function in muscle diseases is the direct link for *MAPRE3* to distal hereditary motor neuronopathy type 7 [79].

It should be mentioned that ADNP plays a dual role, one cytoplasmic with strong microtubule-autophagy-linked activities [13,15,23,26,27,80] and one as a transcription factor, chromatin remodeler [9,16,17]. Our previous gene array and RNA-seq results did not emphasize the ADNP regulation of the muscle disease genes studied here [9,25,80]. Indeed, some regulation may occur, with ADNP interacting with CCCTC-binding factor (CTCF) sites [19], enriched in many genes [81]. However, the low correlations observed at the transcriptomic level (Figure 3) contrasted with the strong data regarding cytoskeletal interactions, imply altered cytoskeletal/autophagy functions as a consequence of muscle gene mutation. Together, these protein interactions may affect *ADNP* transcript cytoplasmic/nuclear content, with ADNP regulating its own transcript [9,76], predictive of motor disease.

Interestingly, cytoskeletal reorganization plays an important role in stretch-induced gene expression, further explaining the effects of ADNP/NAP on similar gene transcript expression, including the muscle-regulating *Myl2* gene [82].

Our current data suggest decreases in mouse male *Adnp* levels with aging in the gastrocnemius lower hind limb muscle/leg (Figure 4). Our previous data looking at the human vastus lateralis (higher hind limb muscle) suggested a modest increase in *ADNP* with aging as well as in the female biceps brachii (one of the main muscles of the upper arm) [13]. Importantly, in our current data (Figure 2) looking at diseases in which increases in ADNP were disease predictive, measurements were performed only in the vastus lateralis, correlating with our previous results in aging [13].

Additionally, the ADNP regulating peptide PACAP [12] was also shown to regulate muscle function in protection against outcome measures in a model of spinobulbar muscular atrophy [83]. Our most recent results also indicated Adnp regulation of the *Adcyap1r1* transcript, encoding the PACAP-specific, PAC1 receptor with a significant increase in bladder *Adcyap1r1* seen in *Adnp^+/−^* versus *Adnp^+/+^* females and correction of the female *Adnp^+/−^* levels to the *Adnp^+/+^* levels by NAP treatment [10]. These findings suggest a cross-regulation of ADNP and PACAP functions, with the PAC1 receptor antagonist, PACAP(6–38), reducing urinary bladder frequency and pelvic sensitivity in mice exposed to repeated variate stress [48] and with the majority of ADNP syndrome patients suffering from bladder training delay [5]. Furthermore, PACAP ameliorates *Adnp^+/−^*-deficiencies that are exacerbated by stress [12].

Together, our results suggest that ADNP plays a key role in muscle, as previously shown for the brain. For example, our previous data showed a reduction in brain ADNP expression in aging animals, exhibiting microtubule-tau pathology (a mouse model for frontotemporal dementia) [84], and a positive correlation between ADNP serum concentrations and IQ test performance in elderly individuals [85]. Indeed, brain-muscle connections were also described before for the dystrophin gene [86], further extended to the autistic ADNP syndrome, with 96–100% of ADNP syndrome children suffering from motor impairments [5,8,87] and with motor impairments correlated with intellectual disability in ADNP cases [7]. Moreover, previous findings linked dysregulation of *ADNP* with aberrant synaptic function in neuropsychiatric diseases [88], possibly playing a part in NMJs, as shown here. Interestingly, sexual differences were found in ADNP expression in Alzheimer’s disease (lymphocytes) [85] and in schizophrenia (post mortem brains and lymphocytes) [27,88].

Our single-cell analysis of the mouse data added an important dimension to the understanding of muscle diseases and ADNP potential involvement in these diseases, implicating additional tissues including, but not limited to adipose tissue stem cells, the pancreas, large intestine, and lung cells with potential multisystem effects.

However, correlation is not necessarily causation. To directly assess the effect of reduction in Adnp expression on motor functions we employed muscle Adnp knockdown in Cas9 expressing mice. Our outcome assessment included the automated CatWalk gait analysis, an exceedingly sensitive tool which allows the identification of an extensive number of gait and locomotion parameters with minimal human interference [89]. The CatWalk paradigm was previously implemented in the assessment of static and dynamic gait parameters in a variety of nerve injury models [90,91,92] including muscular dystrophies [90]. Our further results demonstrated, significant sex-dependent differences in CatWalk performance with G68 Adnp-knockdown female mice exhibiting abnormal interlimb coordination, temporal and spatial parameters. Interestingly, not all limbs were equally affected, suggesting a potential imbalance, which may be more emphasized in the female Adnp-knockdown mice.

Like the CatWalk, the treadmill system, assessing maximal endurance in mice, is usually implemented in models of neurodegenerative disorders including ALS [93], Huntington’s disease [94,95] and Parkinson’s disease [96,97] after therapeutic interventions [98,99], aiming at studying the potential roles of specific genes on muscle function [100,101]. This is a simple, sensitive, and objective test yielding high-throughput detection of endurance abnormalities [102]. Adnp knockdown G68 males were significantly impaired in treadmill performance, ameliorated by NAP treatment. Despite the abnormalities G68 DD knockdown female mice presented in the CatWalk test, these females were potentially able to significantly compensate treadmill performance defects in terms of speed, compared with G68 DD male mice. Furthermore, NAP treatment of G68 females resulted in reduced speed, potentially associated with the partial balancing of otherwise imbalanced paws, as observed in the CatWalk test (showing different effects for right, left, front and hind paws). Together, these results further emphasized sexual differences and established the G68 DD male mice as having significantly impaired treadmill behavior.

Regarding sexual dichotomy, we have shown here a very high correlation of *Adnp* expression to *Gnrhr*, tightly linked to sexual control. We have previously shown sex differences in ADNP expression in the mouse and the human hippocampus [21] as well as in the mouse hypothalamus [103]. Interestingly, not only in muscles, as seen here, but also in other tissues, Adnp regulates other genes in a sexually-dependent manner in the brain [21,25] as well in the spleen [28], which is also reflected in differential gut microbiota expression [104]. Importantly and pertinent to our results here, Adnp [25,28] and NAP [28] regulate the Y-chromosome gene lysine demethylase 5d (*Kdm5d*), which may be partially linked with cardiac muscle differentiation [105]. As ADNP was linked before to heart development in mice [46] and in humans [5], our current data may also imply sex-dependent ADNP effects in adult/aging heart diseases. Most interestingly, we have shown sexual dichotomy in microtubule dynamics also in association with Adnp expression [25], with ADNP regulating steroid pathways [80], impinging on sex differences in muscle function.

To summarize the discussion above, highlighting points for future investigations, we would like to point out the following interesting, but perhaps conflicting results.

(1) As indicated above, DMD and BMD are allelic disorders caused by the *DMD* (dystrophin) gene mutation. However, changes in the muscle expression level of *ADNP* were divided in opposite sides: downregulated in DMD (complete deletion of the X-linked *DMD* gene) and upregulated in BMD (*DMD* mutations). With the DMD protein [86] being one of the longest proteins known (3685 amino acid residues), DMD and BMD exhibit markedly different phenotypes (see also Appendix A), including opposite regulation of *ADNP* as discovered here, which may be of interest for future investigations.

*MAPRE1*, encoding one of the cytoplasmic targets of ADNP [13], was increased in muscles of diseases with decreased muscle *ADNP* (Pompe, DM2, TMD, DMD, Figure 2). However, increases in muscle *MAPRE1* were also observed for BMD and ALS, diseases showing increases in *ADNP* (Figure 2), implicating additional players for further investigation. Similarly, while one may suggest that there is no difference in age and sex between TMD and ALS, these are two very different diseases, with TMD being a late-onset, autosomal dominant distal myopathy, resulting from mutations in the two last domains of titin (associated with the unfolded protein response and altered autophagy) and with ALS being essentially a sporadic disease associated with several genetic deficits (Appendix A). Here, *ADNP* and *MAPRE1* showed a negative correlation in TMD and a positive correlation (increases) in ALS, again suggesting additional mechanisms requiring further investigations.

(2) Our previous data showed that the expression level of human male *ADNP* increased with age in vastus lateralis and biceps brachii and similar age-dependent increases were observed in the male mouse gastrocnemius muscle up to 8 months of age [10]. Here, looking at 18- and 24-month-old mice compared to 3-month-old mice, at the single-cell level, a decrease in the very old age in the male mouse muscle limbs was observed (Figure 4). These results imply that at a very old age, *ADNP* may also decrease in the human muscle.

(3) In the results of the *Adnp^+/−^* mice (Figure 5 and Figure 6), there was an age-dependent difference in terms of genotype related muscle *Adnp* expression, significantly reducing in the 7-month-old male (but not female) mice, with *Myl2* correlating with the hanging wire behavior only in females. In contrast, in 19–27-day-old mice, muscle *Adnp* was significantly decreased in both *Adnp^+/−^* males and females, regardless, *Myl2* was again only decreased in *Adnp^+/−^* females and corrected by NAP treatment. At this young muscle age, *Adnp^+/+^* control males expressed higher *Adnp* and lower *Myl2* concentrations than females, suggesting age and sex-dependent regulation as was noted before in the brain and the spleen of these mice [25,28].

(4) Similar to the *Adnp^+/−^* mice, behavioral outcomes of the genetically edited mice were affected differently by gender (Figure 7 and Figure 8). Abnormalities were found in the treadmill test in males and in the CatWalk test in females. While treadmill performance was improved by NAP treatment in males, it decreased in females in the genome-edited mice. However, in the CatWalk test, NAP significantly improved the Adnp knock-down mouse behavior.

Together with the critical points above, our study limitations include the inability to look at all human different muscles in health and all muscle diseases, to identify a precise pattern and specificity. The strength of our findings resides in the emphasis of ADNP’s crucial role in muscle function, coupled to its binding partners MAPRE1 (EB1) and MAPRE3 (EB3) and the clinical significance of amelioration by NAP (CP201, davunentide) treatment, with marked sexual dichotomy. In this respect, davunetide was tested before in progressive supranuclear palsy (PSP) affecting muscle function and while presenting a clean toxicology profile, it did not present efficacy [106]. However, in this previous PSP clinical study, a pure 4 repeat tauopathy, which may not entirely fit the NAP mechanism of action [24], males and females were mixed and not analyzed separately, which may have further skewed the results. Given our discovered predictive value of *ADNP* levels on muscle function and NAP ameliorative effects in the preclinical setting, future clinical trials should take advantage of individualized precision, personalized medicine with CP201 (davunetide), and pipeline products [25], as well as ADNP regulating neuropeptides [83], presenting potential adjuvant therapeutics.

## Figures and Tables

**Figure 1 cells-09-02320-f001:**
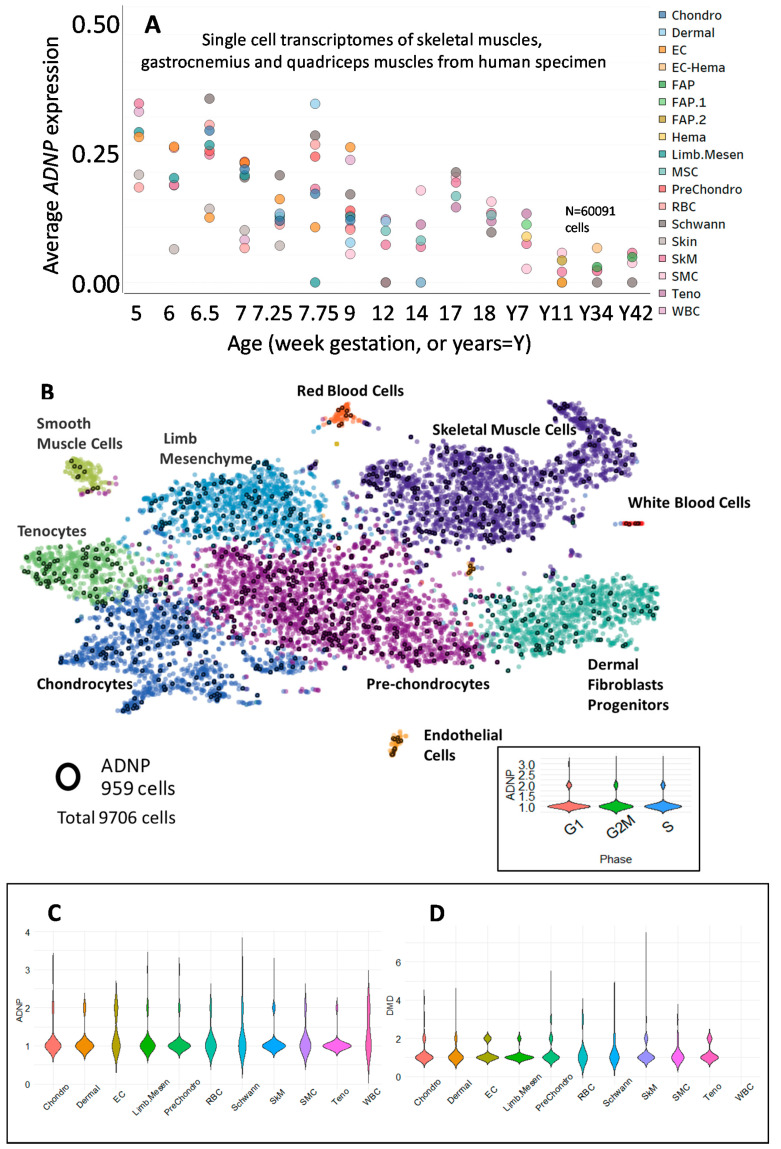
ADNP is expressed in the single human muscle cell. Single-cell transcriptomes of skeletal muscles, gastrocnemius, and quadriceps muscles from a human specimen (GEO dataset GSE147457). (**A**) Average expression levels of ADNP for each cell type and through developmental weeks 5–18 and years 7, 11, 34 42. *n* = 60,091 cells. (**B**) ADNP-expressing cells are marked in black-2D plots of single cells, visualized by the t-SNE algorithm, UCSC Cell Browser [40]. Inset: Violin plot of ADNP expression levels in single cells in phases G1, G2M, and S. *n* = 9706 cells. (**C**,**D**) Violin plots of single-cell expression levels of ADNP and DMD in each cell type. Chondro = Chondrocytes, Dermal = Dermal Fibroblasts Progenitors, EC = Endothelial Cells, EC-Hema = Endothelial, and Hematopoietic Cells, FAP = Fibro-adipogenic Progenitor, Hema = Hematopoietic Cells, Limb. Mesen = Limb Mesenchyme, MSC = Mesenchymal Stromal Cells, PreChondro = Pre-chondrocytes, RBC = Red Blood Cells, Schwann, SkM = Skeletal Muscle Cells, SMC = Smooth Muscle Cells, Teno = Tenocytes, WBC = White Blood Cells [39].

**Figure 2 cells-09-02320-f002:**
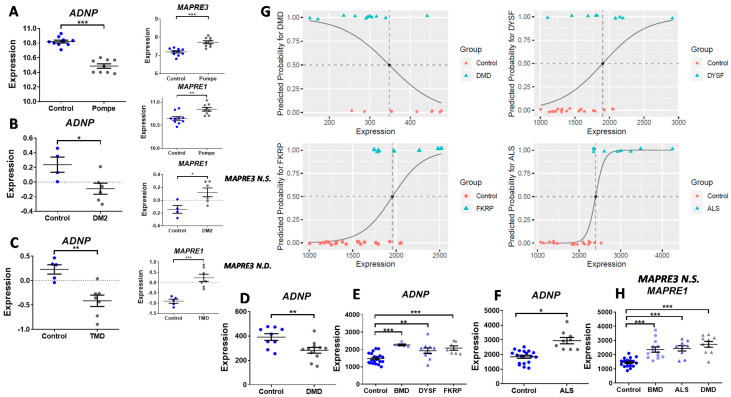
*ADNP* is dysregulated in different neuromuscular disorders. (**A**–**D**) *ADNP* was significantly downregulated in various muscle types: bicep, quadriceps, vastus lateralis and distal muscles from patients with a range of neuromuscular disorders: Pompe *n* = 9 (control *n* = 10, GSE38680), DMD *n* = 11 (control *n* = 11, GSE1007), DM2 *n* = 6 (control *n* = 4, GSE45331) and TMD *n* = 7 (control *n* = 5, GSE42806), as revealed by two-tailed Student’s t-test (* *p* < 0.05, ** *p* < 0.01, *** *p* < 0.001, and ** *p* < 0.01, respectively). (**A**–**C**) *MAPRE3* was upregulated in Pompe disease and *MAPRE1* was upregulated in Pompe, DM2 and TMD, in the same subjects/tissues as above (N.S. insignificant, N.D. not determined). (**E**,**F**) *ADNP* was significantly upregulated in microarray expression levels of vastus lateralis muscle biopsy specimens from patients with various muscle diseases: control (*n* = 20), BMD (*n* = 5), DYSF (*n* = 10), FKRP (*n* = 7) (data set GSE3307, U133B Array), ALS (*n* = 9) (data set GSE3307, U133A Array). One-way ANOVA followed by a Tukey post hoc test (for GSE3307, U133B Array) or Dunn’s Method (for GSE3307, U133A array, due to Equal Variance Test Failed) was performed using SigmaPlot (* *p* < 0.05, ** *p* < 0.01 and *** *p* < 0.001, respectively). (**G**) Logistic regression was performed to estimate the probability of disease diagnoses via *ADNP* levels. DMD; the model estimates a 1.7% decrease in odds of disease, where OR = 0.983, pv(chisq(DF = 1) > 4.873) = 0.027, CI(95%) = [0.968,0.998], AUC = 0.818. DYSF; the model estimates an increase of one unit in “expression” increases the odds for DYSF by 0.3%, where OR = 1.003, pv(chisq(DF = 1) > 5.604) = 0.018, CI(95%) = [1.001,1.006], AUC = 0.805. FKRP mutation; the model estimates an increase of one unit in expression increases the odds for FKRP by 0.6%, where OR = 1.006, pv(chisq(DF = 1) > 5.799) = 0.016, CI(95%) = [1.001,1.010], AUC = 0.9. ALS; the model reports an increase one unit in expression increases the odds for ALS by 1%, where OR = 1.010, pv(chisq(DF = 1) > 3.403 = 0.065, CI(95%) = [0.999,1.02], AUC = 0.963. (**H**) *MAPRE3* did not change in BMD, ALS and DMD, *MAPRE1* was significantly increased, please see D–F for further description of the samples.

**Figure 3 cells-09-02320-f003:**
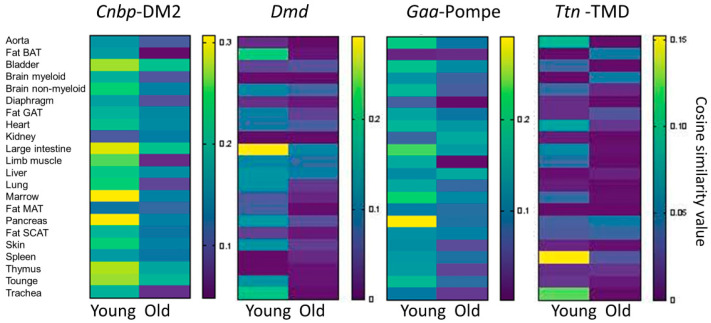
Correlation of *Adnp* and muscle disease genes in the single-cell/general tissue. Heatmaps plotting cosine similarity values (indicating co-expression at the single-cell level) for the genes of interest (indicated above each heatmap) and *Adnp* in young (3 months) and aged mice (18 and 24 months) in 22 tissues (indicated in rows). GAT, SCAT, MAT, and BAT stand for gonadal-, subcutaneous-, mesenteric- and brown-adipose tissue.

**Figure 4 cells-09-02320-f004:**
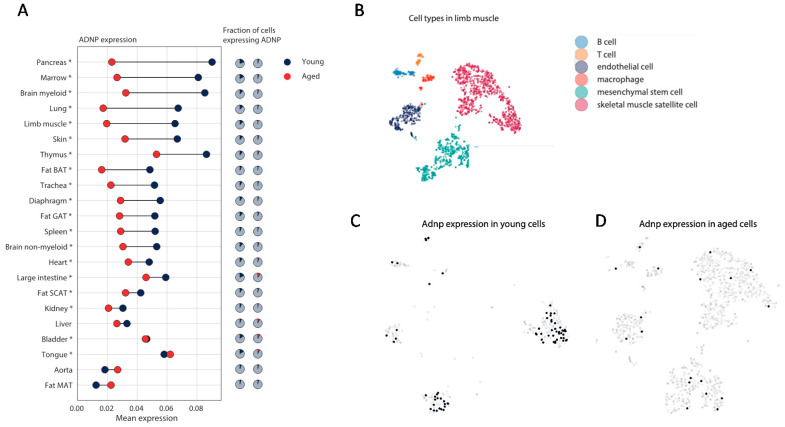
Adnp expression is downregulated with age, specifically in limb muscle cells. (**A**) Mean *Adnp* expression in young (3 months) and aged mice (18 and 24 months) in 22 tissues (indicated in rows). GAT, SCAT, MAT and BAT stand for gonadal-, subcutaneous-, mesenteric- and brown-adipose tissue. Asterisk indicates * *p* < 0.05 using the Wilcoxon rank-sum test. (**B**) Annotated cell type clusters in the limb muscle tissue. (**C**) Out of 542 limb muscle cells in the young mouse 70 express *Adnp* (overlaid as dark blue dots). (**D**) Out of 1315 limb muscle cells in the aged mouse 18 express *Adnp* (overlaid as light blue dots).

**Figure 5 cells-09-02320-f005:**
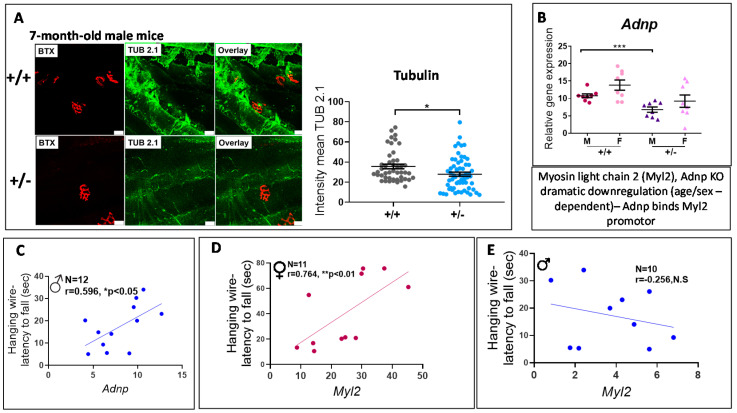
NMJ disruption in the gastrocnemius muscle of *Adnp*-deficient mice correlates behavior. (**A**) Representative whole-mount NMJ immunostaining of 7-month-old (*Adnp^+/+^* CB *n* = 4; *Adnp^+/−^* CB *n* = 4) male mice. The post-synaptic marker nicotinic acetylcholine receptor was labeled by bungarotoxin (BTX, red) and the pre-synaptic marker tubulin was labeled by TUB 2.1 (green). Decreased tubulin intensity was observed in *Adnp^+/−^* CB, compared with *Adnp^+/+^* CB, (* *p* < 0.05). The images were acquired by a confocal microscope at ×20 magnification. Scale bar 25 μm. (**B**) qRT-PCR analysis was performed on mRNA extracted from gastrocnemius muscle of 7-month-old male and female mice (Males: *Adnp^+/+^* CB *n* = 4, *Adnp^+/−^* CB *n* = 4; Females: *Adnp^+/+^* CB *n* = 4, *Adnp^+/−^* CB *n* = 4). Results were normalized to hypoxanthine-guanine phosphoribosyltransferase (*Hprt*). In males, an unpaired Student’s t-test revealed significant differences between CB-treated *Adnp^+/+^* and *Adnp^+/–^* mice (*** *p* < 0.001). Significant correlations were observed between behavioral tests (Hanging wire) and gene expression (*Adnp*
**C**, *Myl2*
**D**,**E**) results in 7-month-old mice. Correlative analyses were performed using either the Pearson correlation coefficient method or the Spearman’s rank correlation coefficient if at least one of the data sets was not normally distributed. Male significant correlations are presented in blue and female significant correlations are presented in magenta.

**Figure 6 cells-09-02320-f006:**
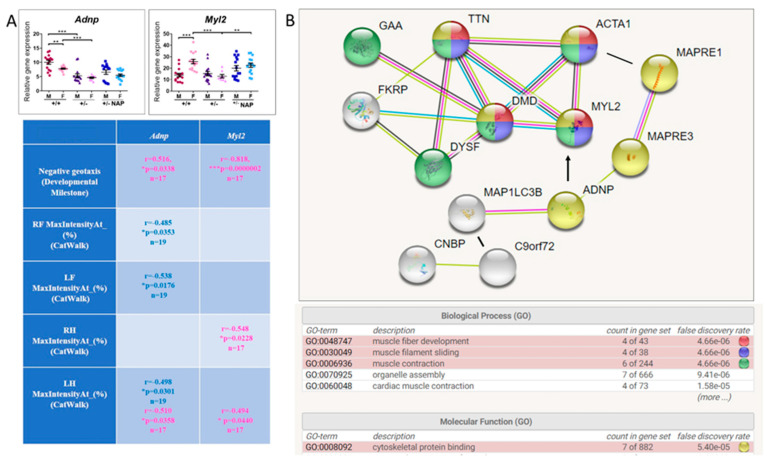
*Adnp* and the Adnp-regulated *Myl2* correlate with muscle function and behavior. (**A**) Gastrocnemius muscle total RNA was extracted from 19-27-day-old mice (males: *Adnp^+/+^ n* = 5, *Adnp^+/–^ n* = 5, *Adnp^+/+^* NAP *n* = 4, *Adnp^+/–^* NAP *n* = 5; females: *Adnp^+/+^ n* = 5, *Adnp*^+/–^
*n* = 3, *Adnp^+/+^* NAP *n* = 4, *Adnp^+/–^* NAP *n* = 5). Results were normalized to *Hprt*. A two-way ANOVA with Tukey’s post hoc test revealed significant differences between vehicle-treated *Adnp^+/+^* and *Adnp^+/–^* mice and between NAP and vehicle-treated *Adnp^+/–^* mice (** *p* < 0.01 and *** *p* < 0.001). Sex differences were determined by unpaired Student’s *t*-test. (**B**) STRING analysis [67] was performed as described in the text, with human muscle disease proteins delineated in Figure 2 and Table 1 and Appendix A. Additional proteins included muscle actin (ACTA1), the ADNP/NAP-binding EB1 (MAPRE1), and EB3 (MAPRE3) and MAP1LC3B as well as the ADNP/NAP-regulated MYL2.

**Figure 7 cells-09-02320-f007:**
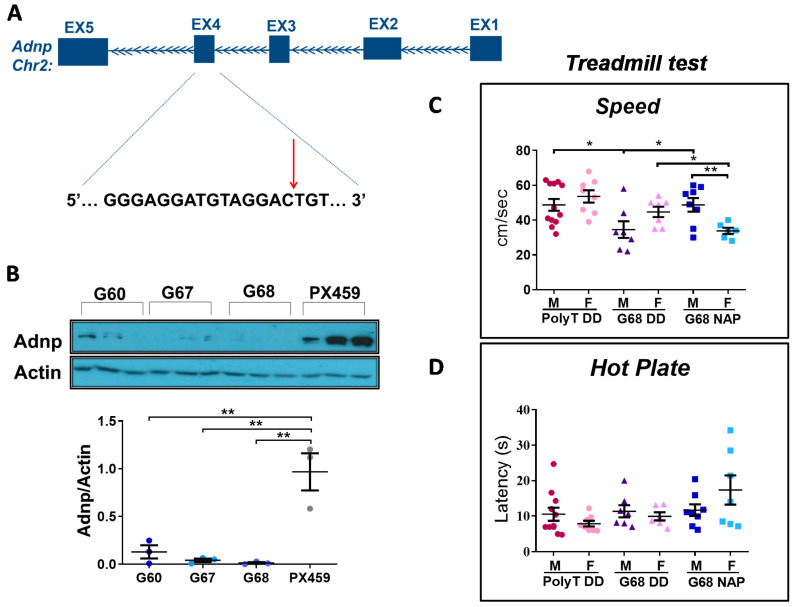
CRISPR/Cas9-mediated knockdown to the *Adnp* gene. (**A**) A schematic representation of the targeted second coding exon (exon No.4) of the mouse *Adnp* gene sequence. The 18-nucleotide sgRNA target sequence is depicted in black, and the red arrowhead indicates the Cas9 cleavage site. (**B**) A successful knockdown of *Adnp* in culture by the CRISPR-Cas9 technology: NIH 3T3 cells were transfected with PX459 plasmid, encoding 3 different *Adnp* sgRNA (termed 60, 67, 68), and an empty plasmid as control (termed PX459), the protein was extracted and subjected to Western blotting (staining with an ADNP antibody and normalized to Actin. One-way ANOVA followed by a Tukey post hoc test revealed that all sgRNA induce a significant knockdown (*n* = 3, ** *p* = 0.002, ** *p* = 0.001, ** *p* = 0.001 for 60, 67, 68 respectively). sgRNA 68 showed the most profound knockdown with a 98.65% reduction in ADNP protein levels, compared with control. All sgRNAs targeted the second coding exon (exon No.4) of ADNP. sgRNA sequences were as follows: 68: 5′-GGGAGGATGTAGGACTGT-3′, 67: 5′-CAGTCCTACATCCTCCCATG-3′, 60: 5′-AACACTACATGGGAGGATGT-3′. (**C**) Treadmill test: A significant reduction in maximum running speed was observed in the G68 male mice group compared to Poly T DD (* *p* < 0.05) and NAP-treated G68 mice (* *p* < 0.05). Poly T (males *n* = 12; females *n* = 8), G68 DD (males *n* = 7; females *n* = 7) and G68 NAP groups (males *n* = 8; females *n* = 7). Furthermore, sex differences were found in the G68 NAP-treated group (** *p* < 0.01). (**D**) No significant differences were observed in the hot plate test. Analysis was performed by using an unpaired Student’s *t*-test. Poly T (males *n* = 12; females *n* = 8), G68 DD (males *n* = 7; females *n* = 7) and G68 NAP groups (males *n* = 8; females *n* = 7).

**Figure 8 cells-09-02320-f008:**
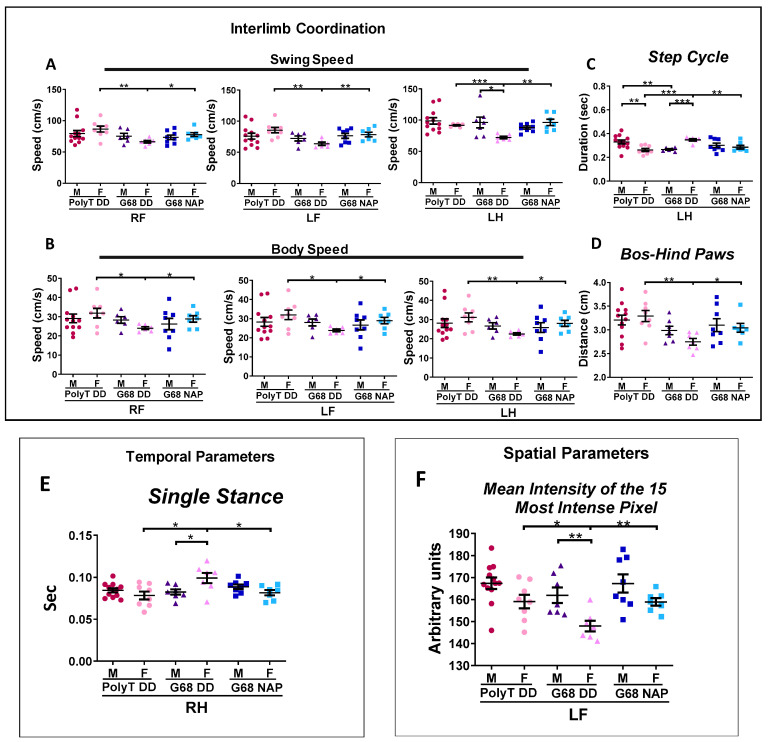
G68 female mice aberrant CatWalk behavior is ameliorated by NAP. CatWalk gait analysis: Poly T (males *n* = 12; females *n* = 8), G68 DD (males *n* = 7; females *n* = 7) and G68 NAP groups (males *n* = 8; females *n* = 7). (**A**) Swing Speed was significantly decreased in G68 female mice, compared with the Poly T DD group, and corrected by NAP treatment: RF (** *p* < 0.01, * *p* < 0.05, respectively); LF (** *p* < 0.01, ** *p* < 0.01, respectively); LH (*** *p* < 0.001, ** *p* < 0.01, respectively). Additionally, sex effect was observed in G68 DD group (* *p* < 0.05). (**B**) Body Speed was significantly decreased in G68 female mice, compared with the Poly T DD group, and corrected by NAP treatment: RF (* *p* < 0.05, * *p* < 0.05, respectively); LF (* *p* < 0.05, * *p* < 0.05, respectively); LH (** *p* < 0.01, * *p* < 0.05, respectively). (**C**) Step cycle an inverse impact was observed in males and females. In G68 DD males, the duration of the step cycle was significantly shorter (** *p* < 0.01), compared with the Poly T DD male group, whereas G68 DD females displayed a significantly longer step cycle, compared with the Poly T DD female group (*** *p* < 0.001), normalized by NAP treatment (** *p* < 0.01). Also, sex differences were found in Poly T DD (** *p* < 0.01) and G68 DD groups (*** *p* < 0.001). (**D**) Base of support (BOS) was significantly smaller in G68 DD female mice, compared with Poly T DD female mice (** *p* < 0.01) and NAP-treated G68 female mice (* *p* < 0.05). (**E**) Single stance was significantly longer in G68 DD female mice, compared with the Poly T DD (* *p* < 0.05) and NAP-treated G68 female mice (* *p* < 0.05). Additionally, sex differences were observed in the G68 DD group (* *p* < 0.05). (**F**) The Mean Intensity of the 15 Most Intense Pixel was significantly decreased in the G68 DD, compared with the Poly T DD female mice (* *p* < 0.05), and NAP-treated G68 female mice (** *p* < 0.01). Sex differences were observed in the G68 DD group (** *p* < 0.01).

**Table 1 cells-09-02320-t001:** *Adnp* correlates with muscle disease gene transcripts at the single-cell level. Cosine similarity values for the genes of interest (see Gene) and *Adnp* in young (3 months) and aged (18 and 24 months) mouse tissue and cell types at the single-cell level, showing an overall loss of correlation with *Adnp* with aging.

	Brown Adipose		Pancreas		Large Intestine		Lung	
*Cell type*	Mesenchymal stem cell	Endothelial cell	Polypeptide cell		Goblet cell	Epithelial cell	bronchial smooth muscle cell	
*Gene*	***Dmd***	***C9orf72***	***Cnbp***	***Fkrp***	***Ttn***	***Gaa***	***Fkrp***	***Dysf***
*Adnp correlation (young)*	0.539	0.643	0.547	0.402	0.392	0.545	0.577	0.383
*Adnp correlation (old)*	0	0	0	0	0	0	0.497	0

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
