# Peer review of "Single Cell ADNP Predictive of Human Muscle Disorders: Mouse Knockdown Results in Muscle Wasting"

_cells, 2020, doi:10.3390/cells9102320_

Round 1

Reviewer 1 Report

The authors have shown that dysregulation of ADNP can be associated with various muscle disorders, and NAP may ameliorate them. The fact that gene editing in gastrocnemius muscle resulted in motor performance alteration is interesting in that it suggests that ADNP is directly involved in muscle disease.

This paper is interesting in that it suggests a link between ADNP and various neuromuscular diseases.

However, the relationship between ADNP changes and their downstream signals and phenotypes varies widely. And It is complicated by the influence of multiple factors such as disease, sex, age, and kind of muscle. I am concerned that it is hard to understand the role of ADNP in muscle disorders without adjusting for the effects of these factors.

Such examples are as follows

The expression level of ADNP increased with age in vastus lateralis and biceps brachii (previous data), but decreased with age in total hind limbs, gastrocnemius, and quadriceps (current data).

Duchenne muscular dystrophy (DMD) and Becker muscular dystrophy (BMD) are allelic disorders caused by dystrophin gene mutation. However changes in the expression level of ADNP were divided in opposite side: downregulated in DMD and upregulated in BMD.

MAPER1, the cytoplasmic target of ADNP, is increased in both diseases with decreased (Pompe, DM2, TMD, DMD) and increased (BMD, DYSF, FKRP, ALS) ADNP.

There is no difference in age and gender between TMD and ALS, but ADNP and MAPR E1 show a negative correlation in TMD and a positive correlation in ALS.

In the results of KO mice (Adnp+/-), there was no gender difference in the effect of ADNP by KO, but Myl2 was decreased only in KO females.

Behavioral outcomes of genetically edited mice are affected differently by gender. Abnormalities were found in the treadmill test in males and in the CatWalk test in females.

NAP treatment significantly increased Myl2 only in females in KO mice, while treadmill performance improved in males but decreased in females in gene-edited mice.

Figure 6 was not included.

Minor points

Line 258, 272, Table 1: “DYSP” is an error of “DYSF”.

Line 271, 272: “Figure 2I” is an error of “Figure 2H”.

Figure 5D: In the text, changes in Adnp mRNA by Adnp KO were significantly observed only in males, and there was no significant difference in females, but Figure 5D shows female Myl2 data. I think they should show male data here.

Line 542: “DMD” is an error of “BMD”.

Line 550:  ADNP expression value of BMD patients did not intersect with the control value in the results.

Author Response

Please see the attachment, in addition, minor spell check was conducted.

Reviewer 2 Report

Kapitansky et al. take multiple approaches, including single cell transcriptomics, RT-PCR, Adnp-deficient mice, CRISPR/Cas9 muscle Adnp knockdown mice, viral vectors, immunohistochemistry, and multiple behavioral tests to determine the involvement of activity-dependent neuroprotective protein (ADNP) in developing human muscle, muscle diseases, and in mutant mouse models. The potential beneficial effects of the ADNP-derived compound NAP in mitigating motor dysfunction in the mouse models was also evaluated.  The main transcriptomics results indicate that ADNP is present in single human muscle cells, including increased expression in embryonic muscle, and that ADNP expression is altered (either downregulated or upregulated) in several different neuromuscular disorders. In global Adnp-deficient mice, disruption of the neuromuscular junction was demonstrated that was correlated with motor behavioral deficits. In adult mice with CRISPR/Cas9-mediated knockdown of Adnp in the gastrocnemius muscle, impairment of motor behavior (gait parameters) was found (mainly in females) that was ameliorated by intranasal NAP administration.

This study is a logical extension of the authors’ previous work on the involvement of ADNP mutations and deficiency with cognitive impairment and global neurodevelopmental delays. The present findings with respect to muscle are novel and interesting and the methods and controls appear appropriate and sound.  Overall, the authors interpret their findings as providing evidence that ADNP plays an important role in muscle development and function and is correlated with certain muscle disorders, likely in a predictive manner. Moreover, they suggest that intranasal NAP treatment may partially ameliorate sex-specific muscle defects in mouse models of Adnp deficiency and knockdown. The authors’ conclusions are supported, for the most part, by the data. The findings connecting muscle deficits with Adnp and beneficial NAP treatment are particularly intriguing. The Discussion is comprehensive and thoughtful and identifies both strengths and limitations of the study. One drawback of the manuscript, at present, is that a figure cited and described in the text is missing from the paper (see below).

The present results may have important implications for the treatment of human myopathies with NAP or other novel ADNP-related agents and targets and, thus, could expand the therapeutic potential of these compounds from brain to muscle disorders. Overall, the present findings should be of substantial interest to the interdisciplinary readership of Cells in general, and to both basic and clinical neuromuscular researchers in particular.   

The following few concerns/suggestions should be addressed to improve the paper:

  1. Unfortunately, Figure 6 is missing from this reviewer’s download of the manuscript, making it impossible to review the data.
  2. In subsection 3.3, at the end of the first paragraph, it appears that the authors made a misstatement when referring to the Fkrp gene and its correlation with Adnp in the aging large intestine epithelial cell. In Table 1 the correlation is listed in the aging bronchial smooth muscle cell. The statement should be corrected. 
  3. Both the second paragraph of subsection 3.4 and the legend to Figure 5 are confusing because they refer to some NAP-treated groups that are not presented in Fig. 5B. This should be clarified.
  4. In Figure S4, the Adnp+/- CB group lists an “n” of only 2, which could be considered borderline for proper statistics. Have the authors since increased the number of mice in this group?

Author Response

Please see the attachment. Minor spell check was conducted, thank you!

Reviewer 3 Report

The manuscript is very well written. Ehe experiments were also outstandingly planned, carried out and documented. There are sufficient amount of citations and background data given to address any further questions regarding the methods and discussion.

The only thing which really needs some improvement are the figures. Some of the figures just show too many data at once. Maybe rearranging and resizing (enlarging) some of the pictures would solve this. Especially Fig 1 B, Fig7 C and D, and lastly the entire Fig 8.

Author Response

Please see the attachment - thank you very much!

Round 2

Reviewer 1 Report

I think the authors responded to the pointed out matters appropriately.

Reviewer 2 Report

The authors have now satisfactorily addressed this reviewer’s previous concerns. Overall, this improved paper should be of considerable interest to the readership of Cells.